# Impact of Preharvest and Postharvest on Color Changes during Convective Drying of Mangoes

**DOI:** 10.3390/foods10030490

**Published:** 2021-02-25

**Authors:** Alioune Diop, Jean-Michel Méot, Mathieu Léchaudel, Frédéric Chiroleu, Nafissatou Diop Ndiaye, Christian Mertz, Mady Cissé, Marc Chillet

**Affiliations:** 1CIRAD, UMR QualiSud, 7, Chemin de l’IRAT, 97410 Saint-Pierre, La Réunion, France; marc.chillet@cirad.fr; 2ITA, Route des Pères Maristes, Hann-Dakar BP-2765, Senegal; ndiop@ita.sn (N.D.N.); christian.mertz@cirad.fr (C.M.); 3CIRAD, UMR QualiSud, 34398 Montpellier, France; jean-michel.meot@cirad.fr; 4QualiSud, Univ Montpellier, CIRAD, Montpellier SupAgro, Univ d’Avignon, Univ de La Réunion, 34398 Montpellier, France; mathieu.lechaudel@cirad.fr (M.L.); frederic.chiroleu@cirad.fr (F.C.); 5CIRAD, UMR QualiSud, 97130 Capesterre-Belle-Eau, Guadeloupe, France; 6CIRAD, UMR PVBMT, 7 Chemin de l’IRAT, 97410 Saint-Pierre, La Réunion, France; 7Laboratoire d’Electrochimie et des Procédés Membranaires (LEPM), ESP-UCAD, Dakar 10200, Senegal; mady.cisse@ucad.edu.sn

**Keywords:** mango, maturity stage, ripening temperature, dried mangoes, color, quality

## Abstract

The purpose of this study was to evaluate the impact of the harvest stage, ripening conditions and maturity on color changes of cv. ‘Cogshall’ and cv. ‘Kent’ variety mangoes during drying. A total of four harvests were undertaken, and the fruits were ripened at 20 and 35 °C for five different ripening times at each temperature. At each ripening time, mangoes were dried at 60 °C/30% RH/1.5 m/s for 5 h. A wide physico-chemical and color variability of fresh and dry pulp was created. The relationships according to the L*, H* and C* coordinates were established using mixed covariance regression models in relation to the above pre- and postharvest (preprocess) parameters. According to the L* coordinate results, browning during drying was not affected by the preprocess parameters. However, dried slices from mangoes ripened at 35 °C exhibited better retention of the initial chroma, and had a greater decrease in hue than dried slices from mangoes ripened at 20 °C. However, fresh mango color, successfully managed by the pre- and postharvest conditions, had more impact on dried mango color than the studied parameters. The preprocess parameters were effective levers for improving fresh mango color, and consequently dried mango color.

## 1. Introduction

Drying represents one of the most common unit operations in the agri-food industry, for product stabilization. It greatly reduces the weight of the products, lowers water activity and slows microbe growth, as well as chemical and bio-chemical reactions. Hence, long storage and transport can be provided at a lower energy cost. Drying also alters the physical chemistry of foods, by causing changes in the mechanical properties [1], aromatic properties [2,3], nutritional properties [4,5], and color properties [6,7,8,9]. Color is generally the top purchase criterion for fruit [10], and for dried mangoes [11] in particular. Natural dried mangoes, without any added sugar or sulfites, are regarded as more dietetic and having a stronger taste [11]. The absence of added antioxidants means that they are prone to browning during preparation, drying and storage. Browning causes a color change in the mango slices from yellow-orange to brown-black, and when over-developed leads to rejection by consumers.

The main fruit browning reactions are both enzymatic and nonenzymatic [12]. Enzymatic browning of mangoes is caused by oxidation of phenolic compounds by polyphenol-oxidases (PPOs) and peroxidases (PODs). This reaction triggers a succession of other reactions, leading to the formation of brown pigments known as melanins [13]. The Maillard reaction, one of the main nonenzymatic browning reactions, is a set of processes between reducing sugars and amino acids, leading to the formation of brown-colored melanoidins [14]. The breakdown of carotenoids, due to the high drying temperatures and oxidative losses due to convective drying [15,16], could also contribute to nonenzymatic browning of mangoes [17].

The agronomic, storage and ripening conditions affect the concentration of browning reaction substrates and of the enzymes catalyzing them. The polyphenols content and PPO activity depends on the variety [18] and increases with mango degree of maturity [19,20,21]. The carotenoids content varies according to the variety [22] and the harvest stage, and increases with the ripening temperature and mango degree of maturity [23,24]. The reducing sugars content increases with maturity [25], but varies according to variety [25,26], agronomic conditions [27] and maturity stage at harvest [28]. These major variations can have a significant effect on the intensity of the various types of browning, and on the color of the product reaching the consumer.

Several authors have studied innovative alternative processes to convective drying, to significantly reduce the browning reactions: conductive multiflash and freeze dying [8], or drying in combination with dual-stage sugar substitution pretreatment [29], or combined hot-air and microwave-vacuum drying [30], assisted by ultrasound [31], low-pressure superheated steam and vacuum drying [17], but without taking into account the pre- and postharvest practices employed on the mangoes.

Knowledge of the effects of preharvest (cropping conditions including irrigation, fruit load and maturity stage at harvest) and postharvest (storage and ripening conditions) on mango pulp color has been little developed [32,33]—and knowledge of the effects of the same factors on color alterations during drying even less so. This type of study has faced the great difficulty or even inability to precisely measure mango maturity, due to the “biological variability” between mangoes from the same batch, however well they may have been selected [34].

The control of color defects is present in the production units as well as in the commercial transactions. It is practiced from the visual comparison of products with reference color charts to instrumental analysis. Chromametry is particularly suitable for measuring mango color as perceived by consumers, which is correlated with quality attributes but also the defects present [35]. A decrease in L*, a* and b* values and an increase in browning index (BI) and total color change (TCC) were associated with deterioration of fruit color during drying [36,37,38]. These color parameters were also used by Zheng [39] to develop a method for the automatic detection of the degree of browning of mangoes with an accuracy of up to 100%. Sturm et al. [40] associated artificial vision system with colorimetric parameters for in-line measurement of color changes (*R* > 0.9). For fresh mango pulp, strong correlations were obtained between the chromatic data (especially the hue angle and a*) and the total carotenoids and beta-carotene content of the pulp [4,15,17,41].

This study was conducted to identify the pre- and postharvest parameters causing color changes during drying. Hence, wide variation in fresh mango pulp composition and color was generated by varying the maturity stage at harvest, the ripening temperatures and times, across two varieties. The relationships between product color before and after drying were analyzed by mixed covariance regression models, taking into account the pre- and postharvest factors studied.

## 2. Materials and Methods

### 2.1. Plant Materials

The experiments were performed on mangoes (*Mangifera indica* L.) of the cv. Cogshall variety in Reunion, and of the cv. Kent variety in Senegal. Two harvests were carried out on Cirad’s experimental orchard at Saint-Pierre (21°19′21.8″ S 55°29′17.9″ E), at the green-mature stage at the start of the harvesting season (early harvest) for Batch 1, and at the yellow-point stage in the middle of the season (late harvest) for Batch 2. Two other harvests in Senegal, on Mr. MBACKE’s orchard, Mboro (15°09′37.6″ N 16°52′13.5″ W), at the green-mature stage at the start of the harvesting season (early harvest) for Batch 3, and at the peak stage (late harvest) for Batch 4. For the Cogshall variety, the maturity stage at harvest was measured using an FMS2 portable fluorimeter (Hansatech, King’s Lynn, UK) on the mango apices [42]. Mangoes with a maximum fluorescence of between 1050 and 1175 correspond to the green-mature stage, while those with a fluorescence of between 750 and 950 correspond to the yellow-point stage [42]. For the Kent variety, the mangoes were harvested by an expert picker, who selected the appropriate mangoes, according to a set of specifications for cargo export, at the green-mature stage. The Batch factor represents the combinations between season time and the maturity stage at harvest, since this space has not been fully explored. On the one hand, the harvest time is a potentially nonlinear variable which we could not reduce to make homogeneous between the two series. On the other hand, the evaluation methods for maturity stage at harvest were different for each variety. Due to insufficient control of these two factors, the four combinations tested were treated as four categories of the same factor.

### 2.2. Mango Ripening and Drying

For each harvest, the mangoes validating the maturity stage at harvest criteria were stored at 20 and 35 °C. Mangoes were taken out after different ripening times. These times were adjusted according to the development (color, texture, and odor) of the fruit, to carry out five withdrawals, the last of which corresponded to very ripe mangoes, on the threshold for fresh consumption. For each withdrawal, five mangoes per ripening temperature category were washed in chlorinated water (0.6% sodium hypochlorite) for 5 min, and rinsed three times in fresh water. After washing, the mangoes were peeled and then sliced using a stainless-steel mandolin slicer 4 mm thick. Four slices from each mango were sampled and dried with a forced convection electric dryer, at 60 °C/30% RH for 5 h with an average air speed of 1 m/s. After sampling four slices, the remainder of the pulp of each mango was ground separately in liquid nitrogen, and then stored at −80 °C for the physico-chemical analyses.

### 2.3. Quality Analysis (Color, Total Soluble Solids, pH, Titratable Acidity)

Twelve color measurements were taken using a chromameter (Minolta CR-400, Tokyo, Japan) on each mango. Three measurements were taken on each mango slice (four slices per mango), before and after drying, on the main axis of the slice; one on the stalk area, one on the central area, and one on the apex area. They were collected in the color space CIE L*, a*, b*. L* is the luminance expressed on a scale ranging from black to white, a* from green to red and b* from blue to yellow. To comment on the color values, the colorimetric space L*, C* (Chroma) and H*(hue angle) was used with the conversion formulae Equations (1) and (2).
(1)$$C^{*}=\;\sqrt{a^{{*}^{2}}+b^{{*}^{2}}}$$
(2)$$H^{*}=\;\tan^{-1}\frac{b^{*}}{a^{*}}$$

The soluble sugars content (TSS) was measured on the defrosted pulp with a portable refractometer (PAL-α, Atago, Tokyo, Japan). The pH and titratable acidity (meq/mL) were measured on the defrosted pulp using an automatic titrator (TitroLine, Schott Instruments, Mainz, Germany). The titration was carried out with a 0.05 N NaOH solution.

### 2.4. Statistical Analysis 

The statistical analyses were carried out with the R software [43]. For each mango, the average of the 12 color measurements was calculated before performing the analyses on the data set. A mixed covariance regression model was built to analyze the effect of the pre- and postharvest factors modulating the relationship between the colorimetric indices (L*, C* and H*) measured on fresh products (independent quantitative variable) and those measured on dry products (dependent quantitative variable). The variety, temperature and their interaction were integrated as fixed effects, and the Batch factor as a random effect, since the harvesters, maturity measurement methods for each variety, and the maturity indicators, varied. Mixed models were then built for each variety with ripening temperature in fixed effect, and the Batch factor in random effect. A deviance test at 5% threshold was performed to test both fixed effects (with Restricted Maximum Likelihood: REML) and random effects on nested models (with Maximum Likelihood: ML). The packages lme4 [44] and MASS [45] were used to build the mixed models, and to compare the models.

## 3. Results and Discussion 

### 3.1. Changes in Quality Criteria of Raw Material According to Variety, Harvest Stage, Ripening Time, and Storage Temperatures

The physico-chemical data of cv. Cogshall and cv. Kent variety mangoes on harvesting and after ripening are presented in (Table 1). The mean mass of the mangoes increased with harvest stage for each variety (significant for Kent). The Kent variety mangoes were on average larger-sized (approximately 600 g) than the Cogshall variety mangoes (447 g), regardless of harvest stage; this agrees with the work of Lebrun et al. [46]. For the harvests at the green-mature stage, there was no significant difference between the total soluble solids content of the two varieties. Our data confirm the results found by Léchaudel and Joas [47], which demonstrate that the Cogshall variety mangoes harvested before 115 days after blooming (DAB corresponding to the green-mature stage) have a total soluble solids content of less than 10. After ripening, the maximum °Brix values (23.5) were attained by the Kent variety. For each variety, the maximum Brix values after ripening were attained by mangoes harvested at later maturity stages (Batches 2 and 4). These mangoes accumulated more metabolites and had a higher quality potential [47]. Batch 2 mangoes appear to have not only accumulated more starch than those from the first one, but their climacteric crisis was triggered before the harvest, causing conversion of starch into soluble sugars and a reduction in citric acid content [47], as indicated by the higher soluble dry extract values and lower titratable acidity values compared to Batch 1.

The mango ripening time to attain a stage corresponding to a very ripe fruit was a direct function of the storage temperature, regardless of the variety and harvest stage (Table 2). The ripening time required to attain this maturity stage decreased with increasing temperature, by approximately 25–30% for a change from 20 to 35 °C. Several studies have shown that increasing ripening temperature leads to a reduction in ripening time [23,48]. The maximum ripening time for the Kent variety was longer than for the Cogshall variety, regardless of the harvest stage and storage temperature studied. This variety, renowned for its exportability [49], confirmed its long potential postharvest lifetime. The ripening time is shorter when the mangoes are harvested at a later stage. In the case of fruit harvested at the green-mature stage, various successive phases have been described; a preclimacteric phase, then the climacteric phase which leads to ripening of the mango [50]. This preclimacteric phase becomes shorter or even nonexistent as the maturity stage at harvest becomes more advanced.

### 3.2. Changes in Color Characteristics of Dried Mango Slices According to Variety, Harvest Stage, Ripening Time, and Storage Temperatures 

As described above for the physico-chemical compositions of fresh mangoes (Table 1), the factor combinations generated wide variation in the color of the dried mango slices (Figure 1). Figure 1a,b are the representations of L* = f (H*) and C* = f (H*), respectively, measured on the dried mango slices from the cv. Cogshall and cv. Kent varieties, and from the different pre- and postharvest factors studied. Changes in these color components varied according to variety, maturity stage at harvest and the ripening temperature.

The distributions of the dots derived from Batches 1 and 2 (Figure 1a,b for cv. Cogshall) exhibit high dispersions compared to Batches 3 and 4 (cv. Kent). This could be due to the greater internal heterogeneity of Cogshall mangoes. Nordey et al. [51] showed that whatever the maturity stage, there were significant color variations in the pulp of the same mango, according to the area where the measurement was taken. In our study, this difference in dispersion of the color indices was found only between the different varieties, but was not affected by the harvest stage, ripening temperature or ripening time.

The L* and H* components were clearly linearly linked (*R*^2^ = 0.78, *p* < 2 × 10^−16^), regardless of preharvest and postharvest conditions (Figure 1a). The same positive correlation was obtained by Gill et al. [32]. Component C* had a typical behavior. As a result, the relationships between C* and H* described crescent shapes, regardless of pre- and postharvest conditions (Figure 1a). These same patterns of relationships between C* and L* (Appendix A) were observed because of the linear relationship between L* and H*. The color varies from a yellow-pale green color (i.e., L* ≃ 85; C*: Cogshall ≃ 50, Kent ≃ 30; H* ≃ 95°) to an orange-chestnut to darker chestnut (i.e., L*: Cogshall ≃ 55, Kent ≃ 64; C*: Cogshall ≃ 80, Kent ≃ 75; H*: Cogshall ≃ 77°, Kent ≃ 73°, (Figure 1a)). The same distribution shape as Figure 1b was obtained by Penchaiya et al. [34], linking a* and b*, which were, respectively, positively correlated with H* and with C*. Each variety had a specific characteristic distribution. For Kent, in the first phase where the dried mangoes were derived from immature mangoes (H* > 95°, L* > 85 and C* < 50; Figure 1a,b), C* underwent a big increase, while H* and L* exhibited little variation, remaining high. The yellow color became more saturated, while H* and L* did not see much reduction, with a slight decrease in green to green-yellow. Thereafter, C* saw very little reduction, while H* and L* saw big falls, changing from yellow to dark orange. These color change mechanisms during ripening were the same as for the varieties cv. Langra [32], cv. Manila and cv. Ataulfo [52], with nearly the same colorimetric index values throughout ripening. The first phase was shorter for the Cogshall variety than for the Kent variety. The ripening and color changes were faster for cv. Cogshall than cv. Kent (Table 1). Hence this first change phase from light green to yellow–dark yellow was not detected due to lack of sampling at the very start of ripening, especially for Batch 1, ripened at 20 °C (day 7). Regardless of the variety, a late harvest or ripening at 35 °C, or these two factors in combination, yielded lower H* and L* and higher C*. The color of these dried pulps appeared to be more orange and purer, which could be appealing to consumers. According to Baloch and Bibi [23], mangoes harvested at an advanced stage and ripened at high temperature have a more attractive color and better sensory rating. The color of the dried Kent slices harvested late and ripened at 35 °C was more orange, characterized by a lower H* min (72°), less intense, with a lower C* max (75) and less dull, with a higher L* min (64) than for Cogshall (with H* min, C* min, and L* min of 76°, 86, 54, respectively). Ripening at 33 ± 2 °C obtained dried mangoes with lower H* values, leading to higher carotenoid and sensory quality contents for Nam Dokmai [4].

### 3.3. Relationships between Color Characteristics of Fresh and Dried Mango Slices According to Variety, Harvest Stage, Ripening Time, and Storage Temperatures

Figure 2, Figure 3 and Figure 4 represent the relationships between the colorimetric indices measured on the dried slices (L* dried, H* dried, C* dried) and those measured on the same slices when fresh (L* fresh, H* fresh, C* fresh), according to various pre- and postharvest factors. All the points are close to the first bisector, which indicates that drying makes little difference to the color of the mango slices, regardless of their history before drying. However, the slopes and intercepts exhibit some variations. The results of the mixed model can be used to test if the variety, harvest stage and ripening temperature significantly affect the color changes caused by drying (Table 3). The factors significantly affecting the relationship parameters vary according to the index studied.

#### 3.3.1. Luminance (L*)

Figure 2 represents the lightness L* of the dried slices as a function of that of the fresh slices, according to the various factors studied. The deviance tests with a threshold of 5% on the mixed models show that the variety and ripening temperature have a very low-significance effect (*p* = 0.04) on the slopes of the relationship between the L* of the dried slices and that of the fresh slices. Conversely, the variety has a very highly significant effect (*p* = 9 × 10^−13^) on the intercepts of these relationships. This is primarily due to Batch 1, ripened at 20 °C, as is confirmed by the highly significant effect of the Batch factor (*p* = 0.003) in the deviance test on embedded models. The first late sampling for this category day 7, (Table 2) meant that high L* values were not obtained. Lower lightness values were obtained for ripening at 35 °C, hence the increase in the domain of variation of L* improved the prediction quality. The increase in ripening temperature obtained lower L* values [32]. Consequently, the points from Batch 1 (20 °C) form a cloud grouped at intermediate L* values (Figure 2); the prediction equation is not suitable for this category, and disrupts the overall model. The result of the mixed regression model for each variety confirms this explanation, since the Batch factor has a significant influence only on the Cogshall variety (*p* = 0.01). For the Kent variety, no parameter has any effect on changes in lightness during drying. The effect on lightness is practically the same regardless of the process followed for drying as implemented.

The formation of brown compounds from the browning mechanisms during drying dulls the fruit slices, and lowers L* [53,54]. The results of Corzo and Álvarez [55] point to a slope of more than 1 for the curves of L* dried = f(L* fresh), i.e., a greater fall in luminance during drying for ripe fresh products characterized by low L*, comparative to immature fresh products with high L*. During ripening, increases in the polyphenols content [21] and PPO activity [19] on the one hand, as well as the reducing sugars content [25] and proteins content [24] on the other hand, should lead to ripe mangoes with a greater browning tendency because of a larger quantity of substrates and enzymes favoring enzymatic browning reactions and the Maillard reaction. Our results on the effect of pre- and postharvest on luminance changes during drying are consistent, since the drying conditions used were unfavorable for the development of enzymatic browning and the Maillard reaction. The constant-rate drying phase was short, with a product surface temperature of around 24 °C, a wet air temperature of 60 °C and 30% RH.

#### 3.3.2. Hue Angle (H*)

Figure 3 represents the hue of the dried slices as a function of the hue of the fresh slices, according to variety, batch and ripening temperature. As for lightness, the regression lines were very close to the first bisector, indicating that drying has only caused very slight hue variations, with slopes very close to 1. The results of the mixed model show that ripening temperature alone had a significant impact *P* = 3 × 10^−3^; (Table 3) on hue changes during drying.

For Batches 1 (Cogshall), 3 and 4 (Kent), harvested at the green mature stage, mangoes ripened at 35 °C had higher slopes and lower intercepts than those at 20 °C. A higher slope and lower intercept indicate that the decrease in H* during drying was greater for ripe mangoes (low H*) than for immature mangoes (high H*). Hence, ripe fruits ripened at 35 °C had a greater tendency for a reduction in hue than ripe fruits ripened at 20 °C during drying. Conversely, the regression line from Batch 2 ripened at 35 °C had a lower slope and a higher intercept than those at 20 °C. At the yellow-point stage, the higher the ripening temperature, the more the color of the mango slices was retained during drying. This specificity should be derived from the yellow-point harvest stage (cv. Cogshall); the fruits have accumulated more metabolites than those harvested earlier, but have also already begun their climacteric crisis favoring all the biochemical mango ripening processes. Joas et al. [56] showed that fruits harvested at the yellow point stage (127–130 days after full bloom) and then ripened at 20 °C had a higher carotenoids content than those harvested at the green stages (106 and 120 days after full bloom) and ripened at 20 °C.

At a given ripening temperature, a later harvest for Cogshall (Batch 2) or for Kent (Batch 4) also resulted in higher slopes and lower intercepts than for mangoes harvested earlier (Batches 1 and 4). These results are directly linked with Figure 1, which also show that the dried slices from mangoes ripened at 35 °C or harvested later (Batches 2 and 4) have weaker hues than those ripened at 20 °C or harvested earlier (Batches 1 and 3). The carotenoids content, which is negatively correlated with the flesh hue [57], increases significantly with harvest stage and ripening temperature [4,23]. The higher carotenoid contents on ripe mangoes should lead to more oxidative reactions and breakdown reactions reducing the hue after drying. For Mahayothee et al. [4], oxidation during convective drying caused significantly greater browning of very ripe mangoes, characterized by a reduction in the H*dried/H*fresh ratio as the mango degree of maturity increased. Our data are consistent with this result, since the H*dried/H*fresh ratio fell with maturity.

#### 3.3.3. Chroma (C*)

The mixed model shows that all the pre- and postharvest factors had a highly significant effect on the slope of the relationship between C*fresh and C*dried (Table 3; Figure 4). The changes to C* during ripening and drying differ according to the variety (Figure 1 and Figure 4). For cv. Cogshall, the mangoes started at an already high C* saturation (dark yellow), which remained practically constant for most categories (see Section 3.2). The low C* variations degraded the prediction quality for cv. Cogshall. Therefore, for this experiment, C* is not a good color tracking indicator during drying for cv. Cogshall, since the prediction models are not robust (*R*^2^ < 0.3 for 20 °C). Rosalie et al. [57] found that for Cogshall, Chroma C* had a weak correlation with carotenoids content, which was more correlated with H*. For this variety, a Floridian type, which is rather orange-colored when ripe, several authors have used hue as the color tracking indicator [42,50,58]. However, for polyembryonic South-East Asian varieties, such as cv. Namdokmai Sithong, the pulp has a pale yellow color on harvesting, which intensifies during ripening with an increase in C* to dark yellow at the end of ripening [59], and it has been reported that C* is well correlated with beta-carotenes content.

For Kent, the extended variation range of C* provided a better prediction quality. Analysis of the mixed model applied to cv. Kent shows that the ripening temperature (*p* = 2 × 10^−5^; *p* = 0.003, respectively, the slope and intercept) and the Batch factor (*p* = 0.009) had a significant effect on the relationship between C* before drying and C* after drying. In fact, the prediction models show that ripening at 35 °C reduced the slope values and increased the intercept values of the regression lines in relation to those at 20 °C (Figure 4). The fall in the slope value indicates that drying caused a greater reduction in the C* of ripe mangoes ripened at 20 °C. At 35 °C, the mangoes had more advanced maturity, with C* values not only higher but also better conserved during dying than at 20 °C (see Section 3.2). Corzo and Álvarez [55] also showed that the change in C* during drying in hot air was less for ripe mangoes than for green or half-ripe mangoes. Mangoes ripened at 35 °C attained maximum C* saturation before drying, with a higher hue (maximum carotenoids content at the ready-to-eat stage), while drying led to a slight increase in saturation for mangoes ripened at 20 °C, which had a lower hue and saturation.

## 4. Conclusions

Drying slices of 4 mm thick at 60 °C, 30% RH, 1 m/s for 5 h proved very good at conserving the color of fresh mango slices. The statistical analyses revealed numerous significant effects between the pre- and postharvest factors and the color alterations occurring during drying, even if their impact was low.

Fruits with advanced maturity, and ripened at 35 °C, were more likely to have a fall in hue H*, but were better at conserving purity of color C* during drying. The fall in luminance L* during drying, indicating browning/darkening, and potentially leading to consumer rejection of the mango slices, proved to be small under our drying conditions, and very little affected by preharvest and postharvest factors. The Kent variety dried slices appeared to have a more attractive color, with a more orange hue.

## Figures and Tables

**Figure 1 foods-10-00490-f001:**
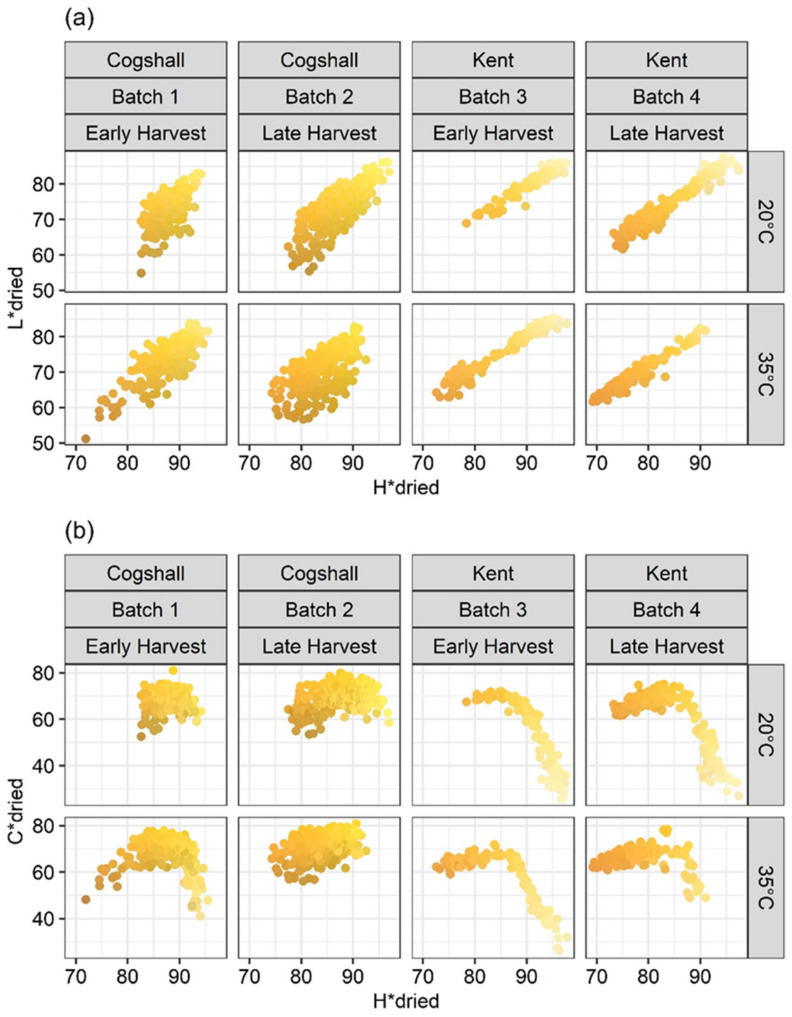
Relationships between L* and H* (**a**) and C* and H* (**b**) from colour measurements on dried mangoes according to varieties, harvest stages, ripening temperatures, and maturity. The colour of the points was obtained by converting L* +, H* and C* coordinates into RGB.

**Figure 2 foods-10-00490-f002:**
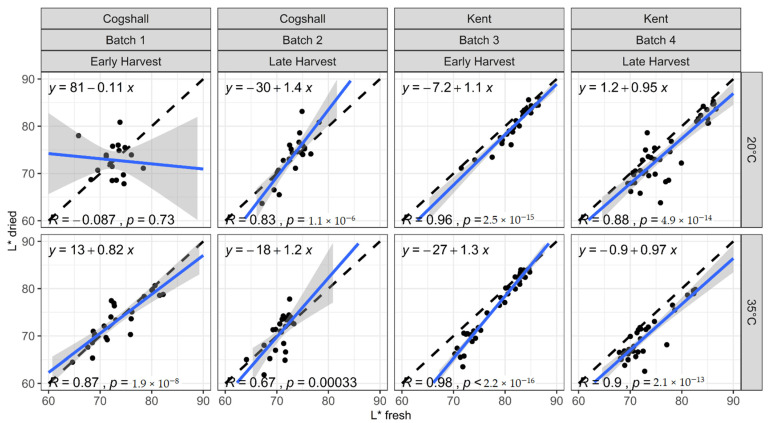
L* measured on dried slices as a function of the same indices measured on the same slices before drying. Black dashed line: first bisector; solid blue line with grey envelope: L*dried linear regression as a function of L* fresh, and 95% confidence interval.

**Figure 3 foods-10-00490-f003:**
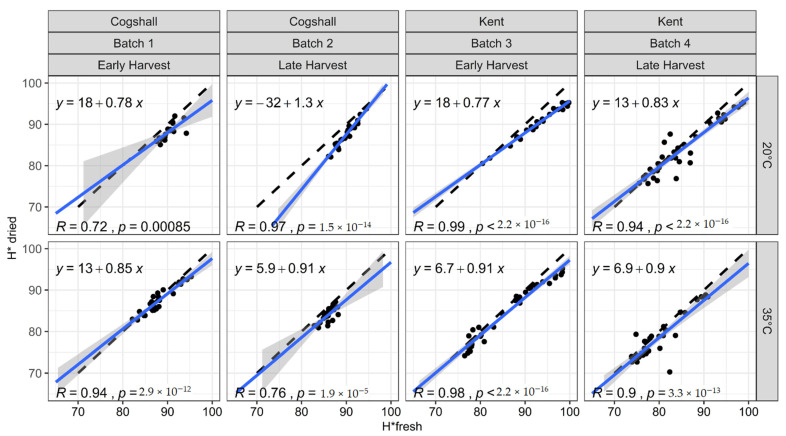
H* measured on dried slices as a function of the same indices measured on the same slices before drying. Black dashed line: first bisector; solid blue line with grey envelope: H*dried linear regression as a function of H*fresh, and 95% confidence interval.

**Figure 4 foods-10-00490-f004:**
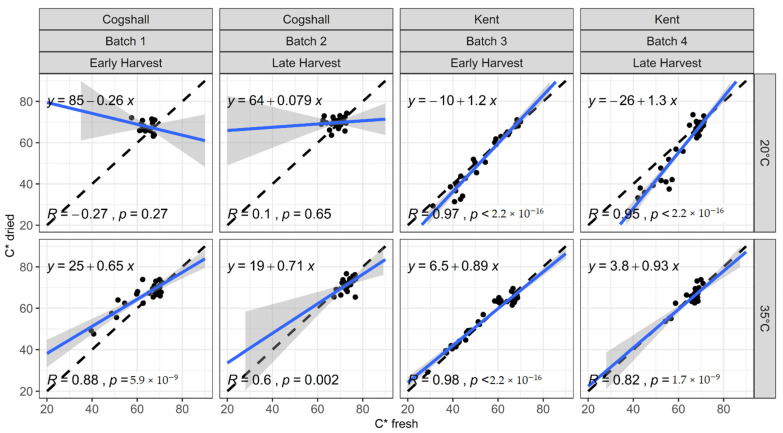
C* measured on dried slices as a function of the same indices measured on the same slices before drying. Black dashed line: first bisector; solid blue line with grey envelope: C*dried linear regression as a function of C* fresh, and 95% confidence interval.

**Table 1 foods-10-00490-t001:** Physico-chemical characteristics at harvest and after ripening of the four mango batches.

Variety	Cogshall					Kent			
Harvest Location	Saint-Pierre, Réunion, France					Mboro, Senegal			
Batch Number	Batch 1			Batch 2		Batch 3		Batch 4	
Harvest date	21 December 2018			1 January 2019		20 June 2019		22 July 2019	
Maturity stage at harvest	Green-Mature (1)			Yellow-point (1)		Green-Mature (2)		Green-Mature (2)	
Harvest stage	Early harvest			Late harvest		Early harvest		Late harvest	
Number of mangoes: at harvest/ripening	6/41			4/47		7/75		7/76	
Mass at harvest (g)	446 (7.3) c			453 (7.5) c		547 (8.1) b		652 (12.7) a	
TSS (°Brix)	At harvest	8.4 (0.5) b		13.7 (2.4) a		6.0 (0.2) b		6.8 (0.1) b	
	Ripening	16.5 (0.3) b		18.2 (0.2) a		14.7 (0.3) c		17 (0.5) ab	
pH	At harvest	3.0 (0.04) b		3.5 (0.14) a		-		-	
	Ripening	3.7 (0.2) b		4.3 (0.2) a		-		-	
TA (meq/100 g)	At harvest	29.5 (1.6) a		15.1 (1.6) b		-		-	
	Ripening	11.8 (2.8) a		6.1 (1.4) b		-		-	
Ripening	Min/max	Min	Max	Min	Max	Min	Max	Min	Max
	TSS (°Brix)	10.7	19.5	14.3	20.7	8.4	19.2	8.1	23.5
	pH	2.85	4.7	3.51	5.46	-	-	-	-
	TA (meq/100 g)	3.8	30.2	2.2	15.2	-	-	-	-

(1) Evaluated using fluorimeter; (2) Evaluated by a professional picker. Values in bracket next to means represent the standard error. The various letters indicate the significant differences (*p* ≤ 0.05) on the means by physico-chemical variable by stage (pairwise comparison test or Fisher Snedecor test).

**Table 2 foods-10-00490-t002:** Ripening time according to variety, harvest stage and ripening temperatures.

Varieties	Batch Numbers	Ripening Temperature (°C)/Sampling Days	2	3	4	5	6	7	8	9	10	11	12	13	14	15	16
Cogshall	1	20 °C						x		x		x		x*			
		35 °C				x	x	x	x	x*							
	2	20 °C				x	x	x	x	x*							
		35 °C	x	x	x	x	x*										
Kent	3	20 °C			x			x		x		x			x		x*
		35 °C			x		x		x		x		x*				
	4	20 °C			x			x		x	x			x		x*	
		35 °C			x		x	x	x	x*							

x* represent the end of the ripening process for each modality (overripe mangoes obtained).

**Table 3 foods-10-00490-t003:** Impact of pre- and postharvest parameters on the slopes and intercepts of the regression lines.

General Mixed Model			L*Dried	H*Dried	C*Dried
(L*/H*/C*Fresh)			<2 × 10^−16^ ***	<2 × 10^−16^ ***	<2 × 10^−16^ ***
Varieties		Slope	0.04 *	0.09	3 × 10^−4^ ***
		Intercept	9 × 10^−13^ ***	0.39	0.67
Ripening Temperature		Slope	0.04 *	3 × 10^−3^ **	0.007 **
		Intercept	0.3	0.7	0.06
Interaction Temperature x Variety		Slope	0.6	0.04 *	2 × 10−6 ***
		Intercept	0.4	0.03 *	0.09
Batches			0.003 **	0.2	0.004 **
**Mixed Model Per Variety**			**L*Dried**	**H*Dried**	**C*Dried**
**(L*/H*/C*Fresh)**			**<2 × 10^−16^ *****	**<2 × 10^−16^ *****	**<2 × 10^−16^ *****
cv. Cogshall	Ripening Temperature	Slope	0.1	0.07	2 × 10^−4^ ***
		Intercept	0.8	0.001 **	0.2
	Batches		0.01 *	0.005 **	1
cv. Kent	Ripening Temperature	Slope	0.05	0.003 **	2 × 10^−5^ ***
		Intercept	0.4	0.3	3 × 10^−3^ ***
	Batches		0.08	0.98	0.009 **

*** Significant at 0.1% threshold; ** significant at 1% threshold; * significant at 5% threshold.

## Data Availability

The data presented in this study are available on request from the corresponding author.

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
