# Peer review of "Impact of Preharvest and Postharvest on Color Changes during Convective Drying of Mangoes"

_foods, 2021, doi:10.3390/foods10030490_

Round 1

Reviewer 1 Report

The harvest stage and ripening condition has verified effect on the textural and chemical parameters of fruit and vegetables, therefore these can have effect on drying efficiency and drying kinetic, respectively. Manuscript foods-1099986 is well structured and it has an interesting topic that has relevance not just for the science but also the practice. Research motivations are clearly defined. Materials and methods are given in details. Manuscript contains interesting findings. Colorimetric based prediction models (regression between pre- and post-harvest factors) are utilizable in the practice. Results are discussed with relevant references.

Comment, suggestions:

I suggest the authors to discuss in more details (in Introduction section) the applicability of colorimetric analysis to detect the quality changes during drying (because colorimetric analysis was the main methods applied in the study)

It it not clearly given how was the parameters of convective drying (temperature, rH, air velocity) choosen?

Reviewer 2 Report

The work seems to be of decent quality for “Food ". However, I have some an important question for the author to include in the manuscript without these results of the manuscript are incomplete in my point of view.

  • The author has not included impact of pre-harvest and post-harvest on Drying on weight loss, carotenoids and total sugar, percentage of acidity and vitamin C. These are important parameters for the quality of the mangoes.
  • Why author did the study only two temperatures 20 °C and 35 °C? Did they go below and above those temperatures?
  • Did author compare with other widely used methods for the drying and how it will affect the cost in long term storage.
  • They have not included figures for the mangoes before and after the drying experiment it would be interesting for the readers
